# The Identification of Novel Therapeutic Biomarkers in Rheumatoid Arthritis: A Combined Bioinformatics and Integrated Multi-Omics Approach

**DOI:** 10.3390/ijms26062757

**Published:** 2025-03-19

**Authors:** Muhammad Hamza Tariq, Dia Advani, Buttia Mohamed Almansoori, Maithah Ebraheim AlSamahi, Maitha Faisal Aldhaheri, Shahad Edyen Alkaabi, Mira Mousa, Nupur Kohli

**Affiliations:** 1Department of Biomedical Engineering and Biotechnology, Khalifa University of Science and Technology, Abu Dhabi 127788, United Arab Emirates; 100063757@ku.ac.ae (M.H.T.); advanidia@gmail.com (D.A.); 100058977@ku.ac.ae (B.M.A.); 100058497@ku.ac.ae (M.E.A.); 100058718@ku.ac.ae (M.F.A.); 100058417@ku.ac.ae (S.E.A.); 2Center for Applied and Translational Genomics (CATG), Mohammed Bin Rashid University of Medicine and Health Sciences, Dubai Health, Dubai 505055, United Arab Emirates; 3Department of Public Health and Epidemiology, Khalifa University of Science and Technology, Abu Dhabi 127788, United Arab Emirates; mira.imousa@ku.ac.ae; 4Center for Biotechnology, Khalifa University of Science and Technology, Abu Dhabi 127788, United Arab Emirates; 5Healthcare Engineering Innovation Group, Khalifa University of Science and Technology, Abu Dhabi 127788, United Arab Emirates

**Keywords:** multi-evidence genes, transcriptomics, epigenomics, network pharmacology

## Abstract

Rheumatoid arthritis (RA) is a multifaceted autoimmune disease that is marked by a complex molecular profile influenced by an array of factors, including genetic, epigenetic, and environmental elements. Despite significant advancements in research, the precise etiology of RA remains elusive, presenting challenges in developing innovative therapeutic markers. This study takes an integrated multi-omics approach to uncover novel therapeutic markers for RA. By analyzing both transcriptomics and epigenomics datasets, we identified common gene candidates that span these two omics levels in patients diagnosed with RA. Remarkably, we discovered eighteen multi-evidence genes (MEGs) that are prevalent across transcriptomics and epigenomics, twelve of which have not been previously linked directly to RA. The bioinformatics analyses of the twelve novel MEGs revealed they are part of tightly interconnected protein–protein interaction networks directly related to RA-associated KEGG pathways and gene ontology terms. Furthermore, these novel MEGs exhibited direct interactions with miRNAs linked to RA, underscoring their critical role in the disease’s pathogenicity. Overall, this comprehensive bioinformatics approach opens avenues for identifying new candidate markers for RA, empowering researchers to validate these markers efficiently through experimental studies. By advancing our understanding of RA, we can pave the way for more effective therapies and improved patient outcomes.

## 1. Introduction

Rheumatoid arthritis (RA) is an autoimmune disease that primarily affects the joints and affects approximately 1–2% of the global population, although its prevalence varies widely by region. In North America and Northern Europe, the prevalence is estimated to be between 0.5% and 1.1%, whereas it is lower in Southern Europe. In Africa, the prevalence ranges from 0.06% to 3.4%. In the Middle East and North Africa (MENA) region, an estimated 120.6 per 100,000 individuals are affected by RA [1,2].

RA is a diverse autoimmune condition characterized by different systemic manifestations; however, continuous inflammation is a key feature of its clinical appearance. This inflammation begins in small peripheral joints and progresses to other joints, resulting in joint damage characterized by cartilage loss and bone erosion [3]. Furthermore, if left untreated, the inflammation also affects extra-articular organs, including the heart, lungs, skin, and the nervous system [4,5]. The pathophysiology of RA is involved with several inflammatory mediators, signaling pathways, and, in particular, regulatory mechanisms such as genetic links, epigenetic effects, and microRNA (miRNA) action. The abnormal activation of the Janus Kinase (*JAK*), Mitogen-Activated Protein Kinase (*MAPK*), NF-kappa B, and phosphoinosis-tide-3-kinase/protein kinase B (*PI3K/AKT*) pathways has been associated with RA [6]. Moreover, the imbalance between pro-inflammatory and anti-inflammatory cytokines and chemokines promotes RA pathogenesis [7]. These, in turn, dysregulate signaling histopathologic pathways and inflammatory mediators, usually as a result of genetic predispositions or environmental factors.

As per the genome-wide association studies (GWAS) catalog, more than 3000 single nucleotide polymorphisms have been associated with RA onset and development. For example, genetic alterations in signal transducer and activator of transcription 4 (*STAT4*), tyrosine phosphatase non-receptor type 22 (*PTPN22*), cytotoxic T-lymphocyte associated protein 4 (*CTLA4*), peptidylarginine deiminases (*PADI*)-2, and *PADI*-4 have been reported to be associated with RA [8]. Similarly, epigenetic changes, including histone modifications, non-coding microRNAs, and DNA methylation, have also been identified as playing a key role in the pathogenesis of RA by regulating the genes associated with RA [9].

The main therapeutic drugs for managing RA include glucocorticoids (GCs), synthetic disease-modifying anti-rheumatic drugs (DMARDs), and non-steroidal anti-inflammatory drugs (NSAIDs) [10]. These drug interventions are designed to alleviate the symptoms and slow the progression of RA. However, a significant number of patients with RA continue to experience suboptimal responses, develop drug resistance, or suffer from severe adverse effects. A key factor contributing to these challenges is the delayed diagnosis of RA, which can hinder early intervention and optimal treatment outcomes [11,12]. Despite significant advances in genetics and epigenetics, the precise molecular signature of RA remains elusive. Identifying novel biomarkers that are yet to be discovered holds promise for improving early diagnosis and optimizing RA management, making this an area of active and high-priority research. Moreover, biomarker discovery provides an opportunity for the targeted and personalized treatment of patients with RA [13].

Recent advances in high-throughput technologies, such as mass spectrometry, microarrays, and next-generation sequencing, have made it easy to recognize biomarkers in different medical conditions by leveraging a multi-omics approach [14]. Multi-omics modalities demonstrate the current state of the art in deciphering the key molecular mechanisms that govern pathology [15]. This is a novel holistic approach to comprehensively analyzing biological systems. It involves integrating information from various “omics” levels, including genomics, epigenomics, transcriptomics, proteomics, and metabolomics, with each level representing a different layer of biological information. Using a multi-omics approach to tackle diseases is a relatively new approach, and only a few studies have confirmed disease activity across multiple omics levels [16,17,18].

In this study, we aimed to integrate information from multi-omics datasets to identify novel therapeutic targets for RA. Advances in computational methods have greatly facilitated the integration of multiple omics levels into biomarker discovery. Bioinformatics-based approaches are increasingly being recognized as essential preliminary steps before wet lab experiments because of their significant benefits, including reduced time, cost savings, and the ability to generate rapid and reproducible results. This study sought to further characterize the molecular signature of RA by integrating transcriptomic and epigenomic data to identify multi-evidence genes (MEGs) for a more comprehensive approach to RA treatment [19,20].

## 2. Results

### 2.1. Dataset Selection

We selected a transcriptomic dataset (GEO accession GSE56649) from 22 subjects, including 13 patients with RA and 9 healthy individuals, and an epigenomic dataset (GEO accession GSE121192) from 16 subjects, including 10 patients with RA and 6 healthy controls, for data integration and candidate marker identification. The sample cohorts were matched based on the biological source used for data collection, specifically peripheral blood.

### 2.2. Integrating Transcriptomics and Epigenomics Datasets for Rheumatoid Arthritis

A total of 94 genes were Differentially Methylated Genes (DMGs) in GSE121192, utilizing logFC values of +1 and −1. Of these, 25 genes were hypermethylated, and 69 genes were hypomethylated. Likewise, GSE56649 showed 2741 Differentially Expressed Genes (DEGs) when using a logFC value of +1 and −1. Of these, 1394 genes were upregulated, and 1426 genes were downregulated. Eighteen genes were found to be shared in the two-way intersect, with statistical significance (*p* < 0.05). A Venn diagram representing these results is shown in Figure 1A. It is evident that out of 18 genes, 13 genes were upregulated: 5 genes were downregulated in the transcriptomics dataset, 5 genes were upregulated, and 13 genes were downregulated in the methylation dataset. The 18 common genes were Nik Related Kinase (*NRK*), FERM Domain Containing 4A (*FRMD4A*), Coagulation Factor III (*F3*), Protein Tyrosine Phosphatase, Non-Receptor Type 13 (*PTPN13*), UDP-Glucose Pyrophosphorylase 2 (*UGP2*), Zinc Finger Protein 542, Pseudogene (*ZNF542P*), Dedicator of Cytokinesis 4 (*DOCK4*), Membrane Associated Guanylate Kinase, WW And PDZ Domain Containing 1 (*MAGI1*), Transforming Growth Factor Beta Regulator 1 (*TBRG1*), Cytochrome C Oxidase Assembly Factor 19 (*COX19*), FYN Proto-Oncogene, Src Family Tyrosine Kinase (*FYN*), Ceroid-Lipofuscinosis, Neuronal 8 (*CLN8*), WD Repeat Domain 1 (*WDR1*), Calmodulin 1 (*CALM1*), Transducin Beta-Like Protein 1 X-Linked (*TBL1X*), Phosphate Cytidylyltransferase 1, Choline, Beta (*PCYT1B*), Small Integral Membrane Protein 14 (*SMIM14*), and Sorting Nexin 3 (*SNX3*). We also analyzed the expression levels of these genes in both datasets. The comparative differential expression patterns of these 18 common genes are presented in the form of a heatmap in Figure 1B, while comparative differential expression patterns of these 18 common genes are presented in the form of a heatmap in Figure 2. Consistent with previous findings, we found an inverse relationship between the 18 genes in the expression and methylation datasets [21]. Details on the 18 MEGs can be retrieved from Appendix A.

Of these 18 common genes, 6 were found in GeneCards, DisGeNET, or both. F3 was the only protein that was found in both databases, while MAGI1, FYN, and PTPN13 were found in GeneCards, and CALM1 and CLN8 were reported in DisGeNET. The rest were not reported in these databases; however, it is unclear whether they are associated with RA. Therefore, their potential as novel candidate markers of RA cannot be ignored.

### 2.3. Protein–Protein Interaction Network

Among the 18 MEGs, *F3*, *PTPN13*, *UGP2*, *DOCK4*, *MAGI1*, *FYN*, *WDR1*, *CALM1*, *TBL1X*, and *SNX3* were associated with Homo sapiens proteins, as shown in the STRING database (Figure 3). MCODE application in Cytoscape showed that this network encompasses three key networks, termed clusters 1, 2, and 3 (Table 1), named in serial order based on the MCODE score. Each clustered protein group was further investigated for BP-, MF-, and CC-related GOs terms. For the GO-BP term, cluster 1 was reported to be primarily involved in signaling pathways such as the PI3K/AKT signaling pathway and the ephrin receptor signaling pathway, while cluster 2 was reported to be involved in biological processes related to glycogen synthesis and metabolism. Similarly, cluster 3 was found to be involved in processes related to cell–cell junctions. For GO-CC, the highest number of proteins in cluster 1 were found to be enriched in the plasma membrane and cytosol, while it was enriched in the cytosol and cytoplasm for cluster 2 and the plasma membrane and bicellular tight junctions for cluster 3. Similarly, for GO-MF, most of the proteins in all three clusters were found to be involved in binding to different proteins (including enzymes). The details of all GO terms concerning these clusters are shown in Figure 4 (Appendix A).

### 2.4. Pathways Analysis

Of the 18 MEGs, 11 genes were involved in one or more KEGG pathways (Figure 5), with CALM1 being enriched in the highest number of pathways. Among these 11 genes, *TBL1X*, *COX19*, *UGP2*, *PCYT1B*, *DOCK4*, and *SNX3* were potential novel genes identified in this study. A full list of pathways found in these MEGs is given in Appendix A, including both RA-related and non-RA-related pathways and references. Among the novel genes, *UGP2* was involved in the highest number of pathways, followed by *PCYT1B*.

### 2.5. Micro-RNA Analysis

In total, 120 microRNAs (miRNAs) were predicted to be related to RA using the miRWalk online server. Similarly, MEGs also demonstrated a regulatory relationship with 61 miRNAs, 32 of which were found to be common with RA-related miRNAs (Figure 6). Out of 11 MEGs, 7 MEGs may potentially serve as novel predicted genes for RA. In fact, *PCYT1B* showed interactions with 15 miRNAs, and *SMIM14* showed interactions with 6 miRNAs reported in RA.

## 3. Discussion

Integrating transcriptomics and epigenomics and combining gene expression data with regulatory information provides a thorough understanding of disease mechanisms. This method facilitates the identification of biomarkers and therapeutic targets by validating the biological significance of MEGs, uncovering regulatory processes that impact diseases, and supporting the development of personalized treatments to better understand disease variability and progression.

The integration of the two omics layers has been previously used to investigate MEGs in different forms of cancers [22], osteoarthritis [23], autoimmune diseases [24], and diabetes and its related complications [25,26]. Similarly, several studies have been conducted to investigate MEGs for RA at different omics levels [27,28,29]. However, only one study (Whitaker et al. in 2015 [30]) combined transcriptomics and epigenomics to report MEGs in RA. This study focused on the integration of three different datasets from synovial tissues. The present study integrates and analyzes multi-omics data retrieved from peripheral blood.

In this study, we identified 18 MEGs, common in both transcriptomic and epigenomic datasets, unraveling their role in association with RA. Among these 18, *FYN* [31], *MAGI1* [32], *PTPN13* [33], *F3* [34], *CLN8*, and *CALM1* have been reported previously to be linked to RA. However, the remaining 12 have not been shown to be associated with RA. Among the 12 genes, 6 (*SNX3*, *TBL1X*, *COX19*, *UGP2*, *PCYT1B*, and *DOC4*) were found to be involved in RA-related KEGG pathways. The *TBL1X* is involved in RA through the sphingolipid signaling pathway, as previous studies linked sphingolipids (SPLs) to RA [35]. During inflammation, the sphingomyelin phosphodiesterase enzyme (SPL-metabolizing enzyme) hydrolyzes sphingomyelin into ceramide (Cer), increasing Cer accumulation, which induces the release of inflammatory factors, leading to the severity of inflammatory diseases, including RA. Another gene, *COX19*, is also involved in thermogenesis. Methotrexate (MTX) is an RA treatment that increases thermogenetic gene expression in brown and beige adipose tissues in mice, which prompts cold resistance, improved glucose homeostasis, reduced hepatosteatosis, and decreased inflammation, providing novel evidence of the role of thermogenesis in inflammatory diseases, including RA [36]. Another piece of evidence states that Irisin (a myokine or adipokine) is associated with thermogenesis, as its low levels in RA lead to increased inflammation and oxidative stress. Irisin is responsible for bone homeostasis and the browning of white adipocytes, which enhances thermogenesis; therefore, its low levels are linked to inflammation and dysregulation of bone homeostasis in RA conditions [37]. Likewise, *UGP2* can also be linked to RA through metabolic pathways, such as pentose and glucuronate interconversions, amino sugar and nucleotide sugar metabolism, and starch and sucrose metabolism. Alterations of these pathways contribute to the development of RA [38]. Similarly, *PCYT1B* is involved in Glycerophospholipid metabolism, in which RA patients experience abnormal metabolism [39]. Additionally, *DOCK4* is part of the RAP1 signaling pathway, where deregulated signaling in RA cells causes persistent free radical production [40]. Another MEG, *SNX3*, was also found to be associated with RA through endocytosis, which is linked with cytokine regulation, antigen presentation [41], and synovial hyperplasia [42]. Of these six genes, four were also demonstrated to regulate various RA signaling pathways through complex PPI. Some of the most enriched ones are the PI3K/AKT signaling pathway and the ephrin receptor signaling pathway [43], which shows the relevance of these MEGs in RA etiology. Other MEGs, including *TBRG1*, *SMIM14*, *FRMD4A*, *ZNF542P*, *NRK*, and *WDR1*, are novel candidates that have not been previously identified in the KEGG pathways or PPI networks. Although there has been no prior research on these genes, their differential expression in both the transcriptome and epigenome of patients with RA suggests that they may play a significant role in RA pathogenesis. Functional studies can be conducted to investigate the specific biological roles of these genes in RA to assess the impact on inflammatory pathways and immune responses, thereby elucidating their contributions to disease mechanisms

The miRNAs are non-coding single-stranded RNAs that regulate gene expression by inhibiting or degrading messenger RNA. These miRNAs have been reported to be associated with different diseases, including RA [44]. In RA pathophysiology, these miRNAs control immune responses, affecting the severity of inflammation and joint damage [45]. The novel MEGs reported in this study were found to have regulatory relationships with RA-related miRNAs. MEGs (*TBRG1*, *SMIM14*, *FRMD4A*, and *NRK*) that did not show any presence in RA-related KEGG pathways or PPI networks were found in this gene-miRNA relationship, showing that they might play a role in RA pathogenesis. Only two novel MEGs, *ZNF542P* and *WDR1*, were not associated with the RA-related KEGG pathways, PPI network, and RA-related miRNA reported to date. Nevertheless, since they are differentially expressed in both the transcriptome and epigenome of patients with RA, they could potentially have an association with RA, which has not yet been explored.

Other previously reported MEGs show promise as a therapeutic target for RA. *F3* has been implicated in inflammatory processes, with elevated levels correlating with thrombotic complications in autoimmune conditions, including RA, where coagulation changes often significantly contribute to disease progression [46,47]. Additionally, *PTPN13* is known to play a role in the T-cell regulation and signaling pathways associated with inflammation, suggesting its potential as a target for RA therapies [48,49,50]. *MAG1*, *FYN*, and *TBRG1* have been investigated in the context of cancer and cellular migration, which may provide insights into immune cell dynamics and the regulation of inflammatory pathways [51,52,53]. As research progresses in elucidating the roles of these genes in RA, further clinical trials will be critical to establish their therapeutic potential and efficacy in managing this complex disease.

In summary, this study explored the integration of the multi-omics datasets to identify potential therapeutic targets for RA. The results of the multi-omics integration resulted in some novel MEGs that have previously not been explored for RA, opening avenues for future research. Genes identified as part of RA-related gene ontologies and KEGG pathways could serve as potential biomarkers for the early diagnosis and monitoring of disease progression. Additionally, since the expression of these genes is altered in RA, they may represent valuable targets for developing treatment strategies. Previous studies have reported similar bioinformatics-based approaches for identifying potential RA biomarkers. For instance, *CCL5* was identified as a possible biomarker for RA diagnosis using bioinformatics and was subsequently found to be differentially expressed in RA patients, validating the accuracy of the methodology used in this study [54,55]. The current study, however, is limited by the use of only two omics modalities. Future studies should incorporate other omics modalities, such as proteomics and metabolomics, that would further unravel the molecular signatures encompassing other potential therapeutic targets. Further studies should also incorporate more datasets with larger sample sizes and validate their relevance in RA pathogenesis using experimental methods. Additionally, the data from the current study should be compared to data from other biological samples, such as synovial tissue or synovial fluid, to examine the differences in candidate marker expression amongst different tissue types related to the same disease.

## 4. Materials and Methods

### 4.1. Data Collection

This study used the Gene Expression Omnibus (GEO) [56] to retrieve publicly available online databases. To identify the datasets relevant to our study, we used the following keywords: “rheumatoid arthritis,” “RA gene expression,” and “RA methylation.” The data selection criteria were as follows: (1) samples must be collected from the same type of biological fluid for disease and control conditions, and (2) samples must be obtained from drug-naïve patients with RA. Cohorts were matched based on the sample type used for collecting the gene expression or methylation data, i.e., peripheral blood. Studies that utilized synovial fluid or synovial tissue as samples for collecting gene expression and methylation data or undergoing treatment were excluded.

### 4.2. Data Processing and Analysis

The GEO2R tool (https://www.ncbi.nlm.nih.gov/geo/, all accessed on 4 October 2024) was used to create volcano plots for data analysis. Force normalization and log transformation were applied to address scale and distributional difficulties and improve the interpretability of the data. Furthermore, the Benjamini and Hochberg (False discovery rate; FDR) method was used to adjust *p*-values obtained from multiple comparisons, controlling the expected proportion of false positives. A detailed assessment of up-regulation and down-regulation using differential expression analysis was conducted. Upregulation is indicated by a positive log fold change, while downregulation is indicated by a negative log fold change. Data filtration was performed using the log fold change and the *p*-value. The initial log fold change thresholds for upregulation and downregulation were set at +1 and −1, respectively. A *p*-value threshold of 0.05 or lower was used to identify statistically significant results. In the graphs, *p*-values were represented as −log10(*p*-value), where a *p*-value of 0.05 corresponds to 1.3. Thus, −log10(*p*-value) ≥ 1.3 was considered statistically significant.

### 4.3. Data Integration and Venn Analysis

Following data analysis, data integration was performed to identify the shared MEGs between the transcriptomics and epigenomics datasets. Initially, UniProt (https://www.uniprot.org/) [57] was used to corroborate standard gene names. The interactiveVENN tool (http://www.interactivenn.net/) [58] was then employed, enabling the comparison of gene names between transcriptomic and epigenomic datasets and displaying the common genes. Finally, GeneCards v.5.21 (https://www.genecards.org/) [59] and DisGeNET v.24.3 (https://disgenet.com/) [60] databases were employed to identify RA-related gene targets in “*homo sapiens*”.

### 4.4. Protein–Protein Interaction Network Analysis

STRING v.12.0 (https://string-db.org/) was used to investigate the protein–protein interactions (PPIs) of MEGs with other proteins of *Homo sapiens*. A list of PPI with a confidence score of ≥0.4 was retrieved and opened in Cytoscape software v.3.10.1 [61]. Following this, the Molecular Complex Detection (MCODE) plug-in v.2.0.3 [62] was employed to identify clusters in the PPI network. For the MCODE-based network scoring function, the false degree cutoff was set to 2, while for cluster finding, the K-core cutoff was also set to 2. Clusters with an MCODE score > 5 were identified and distinguished from other proteins by assigning different colors. The Database for Annotation, Visualization, and Integrated Discovery (DAVID) [63] was used to conduct gene enrichment analysis for gene ontology (GO) terms related to biological processes (BPs), cellular components (CCs), and molecular functions (MFs), with a false discovery rate (FDR) < 0.05 considered statistically significant. Following these GO investigations, the Hiplot online server [64] was used to generate bubble plots to visualize the results.

### 4.5. Pathway Analysis of Multi-Evidence Genes

The DAVID online tool v.2023q4 (https://davidbioinformatics.nih.gov/home.jsp) was also used to investigate the Kyoto Encyclopedia of Genes and Genomes (KEGG) pathways of the common MEGs [65]. Finally, Cytoscape v3.10.1 was chosen to generate a network demonstrating the linkage between MEGs and RA-related KEGG pathways [61].

### 4.6. Gene-miRNA Analysis

The miRWalk v.3 (https://mirwalk.umm.uni-heidelberg.de/) [66] was used to investigate interactions between MEGs and the predicted microRNAs (miRNAs) using TargetScan [67] as the screening database, 3′ Un-Transcribed Region (UTR) was considered as the target gene binding region, and relationship score was set to 0.95. Similarly, miRNAs associated with RA were also found using the same software and the same parameters. Finally, common miRNAs between the two datasets were identified to find out MEGs’ interaction with the RA-associated miRNAs. Cytoscape software v 3.10.1 [61] was employed to build a network for visualization. The overall methodology opted in this study to achieve objectives is shown in schematic diagram in Figure 7.

## 5. Conclusions

In this study, we identified 18 novel MEGs by integrating transcriptomic and epigenomic databases from patients with RA. Among these, six MEGs have been previously reported in the literature, while twelve are newly identified and have not been documented in the context of RA before. Remarkably, ten of these unique MEGs are associated with RA-related KEGG pathways, PPI networks, and/or miRNA interactions, underscoring their potential role in the pathogenesis of RA. Therefore, we propose that these lesser-known MEGs could serve as promising new targets for enhancing our understanding of RA etiology and for developing targeted therapies.

## Figures and Tables

**Figure 1 ijms-26-02757-f001:**
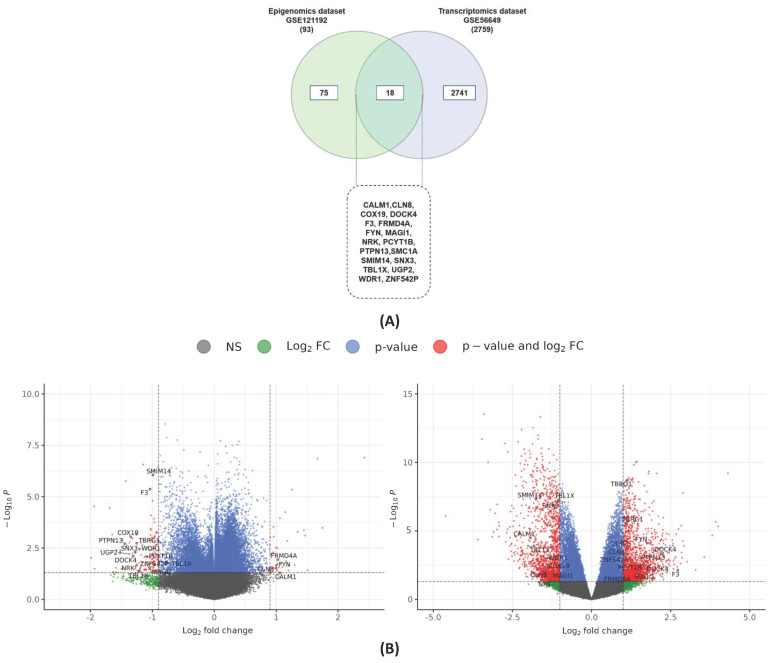
(**A**) Venn diagram showing the intersecting genes between epigenomics and transcriptomics datasets. (**B**) Volcano plots of epigenomics and transcriptomics datasets. The expression difference is considered significant for a log 2-fold change of 1 (outer light gray broken vertical lines) and for a *p* value of 0.05, log(FDR) of 1.3 (broken horizontal line). NS = non-significant, FC = Fold change.

**Figure 2 ijms-26-02757-f002:**
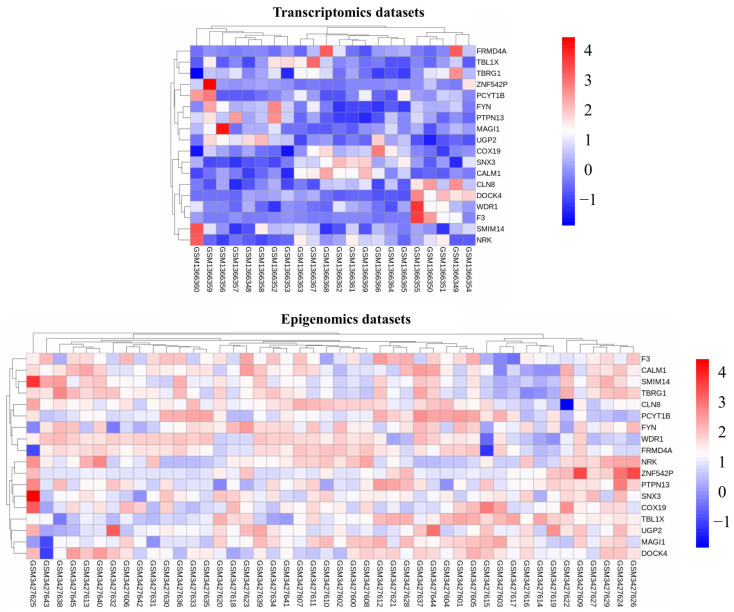
Heat map of the 18 MEGs identified from the transcriptomics and methylation datasets. The color key indicates gene expression levels, with red representing higher expression and blue representing lower expression.

**Figure 3 ijms-26-02757-f003:**
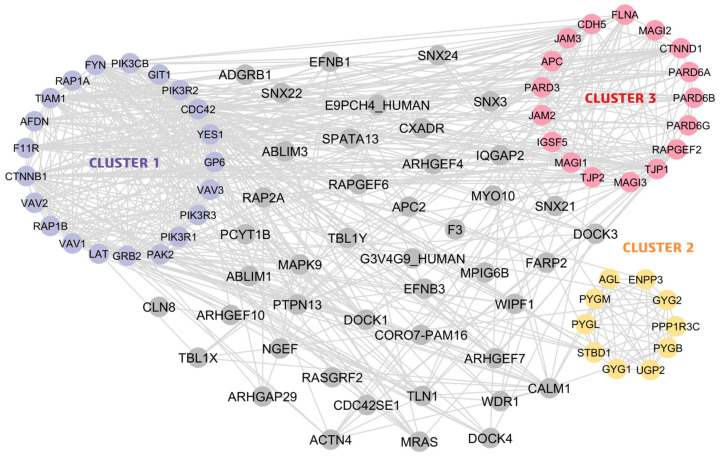
Cytoscape-generated PPI networks of MEGs reported in this study. PPIs are indicated in closely associated clusters and are represented with different colors based on the cluster number. Each node represents a single protein. Cluster 1 is depicted with lavender nodes. Cluster 2 is shown with light orange nodes, and cluster 3 is shown with pink nodes. Nodes in gray represent proteins that do not belong to any of the defined clusters, as identified by the MCODE application in Cytoscape.

**Figure 4 ijms-26-02757-f004:**
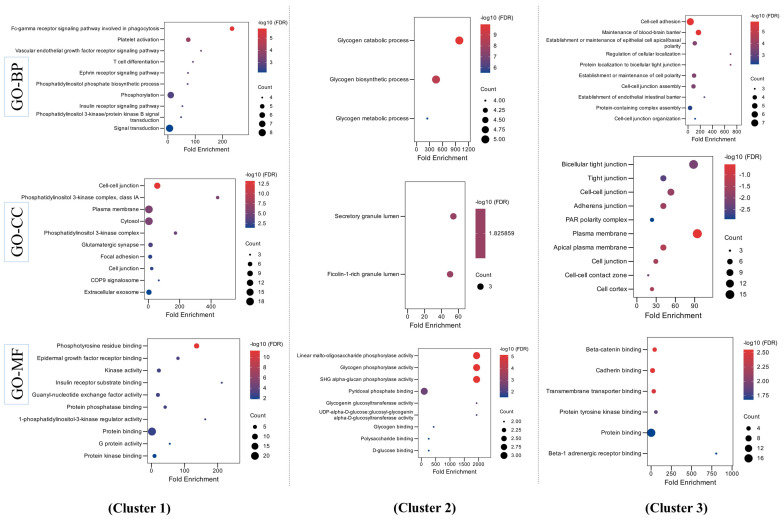
Bubble plots for enriched gene ontology (GO) terms for biological processes (BPs), cellular components (CCs), and molecular functions (MFs) of MEGs in clusters 1–3. The color of the nodes shown with a gradient from red to blue is according to the decreasing order of −logF10(FDR). The size of each node is according to the number of gene counts.

**Figure 5 ijms-26-02757-f005:**
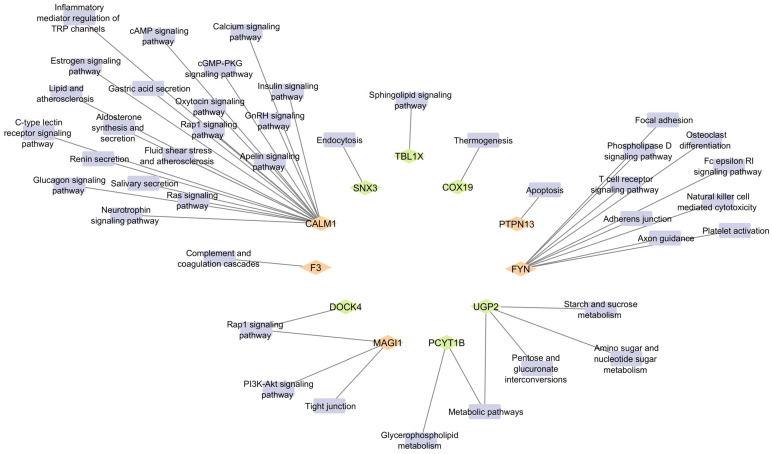
MEGs and their associated RA-related KEGG pathways identified using the DAVID database. Diamond-shaped nodes represent MEGs identified in this study, with green diamonds indicating novel MEGs and orange diamonds indicating previously reported MEGs. Indigo rectangular nodes represent KEGG pathways in which these MEGs are involved.

**Figure 6 ijms-26-02757-f006:**
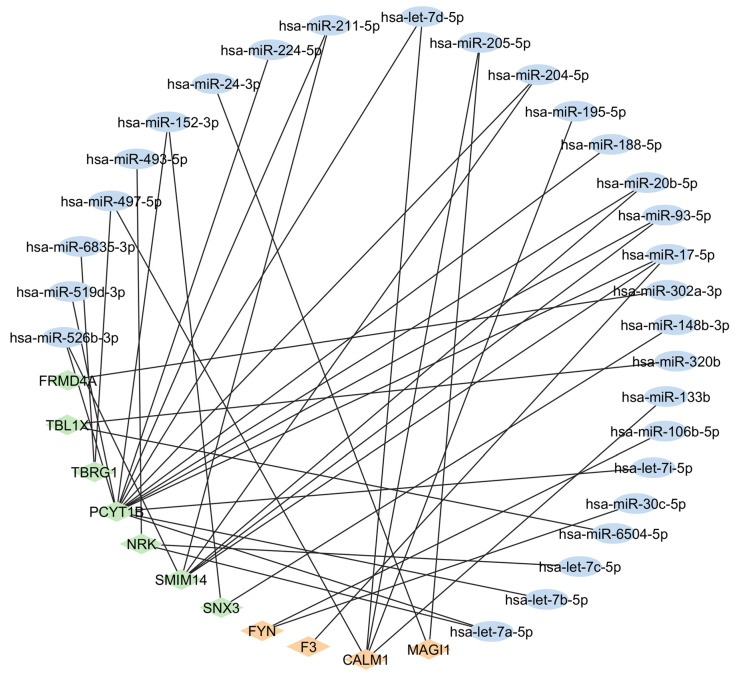
Interaction of MEGs (diamond nodes) with miRNAs (oval nodes) related to RA. Oval-shaped orange color nodes represent miRNA, while orange and green color diamond-shaped nodes show previously reported MEGs and novel MEGs identified in this study, respectively.

**Figure 7 ijms-26-02757-f007:**
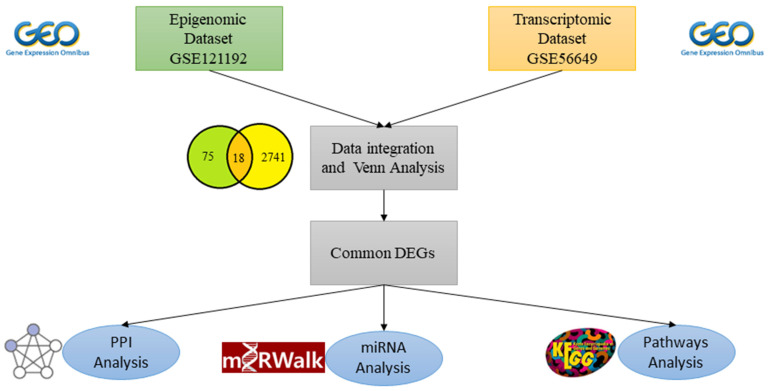
Schematic diagram representing the multi-omics integration and bioinformatics tools used in this study. miRNA refers to microRNA, and PPI refers to protein–protein interaction.

**Table 1 ijms-26-02757-t001:** Details of the clusters found in the PPI network of MEGs, where each node represents one protein, and each edge shows one interaction of one node with another one.

Cluster	Number of Nodes	Number of Edges	MCODE Score
1	17	115	14.375
2	10	42	9.333
3	14	45	6.923

## Data Availability

Data are available as Appendix A.

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
