# Peer review of "The Identification of Novel Therapeutic Biomarkers in Rheumatoid Arthritis: A Combined Bioinformatics and Integrated Multi-Omics Approach"

_ijms, 2025, doi:10.3390/ijms26062757_

Round 1
Reviewer 1 Report
Comments and Suggestions for Authors
By analyzing microarray-based transcriptomic and methylation data, the authors identified eighteen multi-evidence genes (MEGs), 12 of which are novel in RA research. Bioinformatics analysis revealed these genes are part of protein-protein interaction clusters linked to RA-related pathways and interact directly with RA-associated miRNAs, underscoring their potential role in disease progression. The manuscript contains significant design flaws. Below are my comments.
1. The authors' use of "the highest number of common genes" as a criterion for data collection is not well-justified. The methylation dataset (GSE121192) used in the manuscript includes 46 samples across three different cell types, a detail the authors did not address. Given the small sample size (13 patients vs. 9 controls; 10 patients vs. 6 controls), the study's statistical power to identify meaningful genes is limited. Larger datasets, such as GSE15573 and GSE176168, could have been considered to enhance the robustness of the findings. Even though all the datasets involve small sample sizes, methods like meta-analysis could have been employed to improve detection power. Relying on one or two datasets is not an optimal strategy.
2. Have the authors reviewed the original papers that provided the publicly available datasets? The number of differentially expressed genes (DEGs) reported in this manuscript appears to differ significantly from those identified in the original studies. The authors should provide a clear explanation for these discrepancies.
3. The manuscript lacks substantial discoveries or significant findings. Conducting several bioinformatics analyses alone is insufficient for a publication-level scientific paper. Experimental validation of the identified genes is needed to lend greater credibility to the results.
4. Figure 1 is essential, but it is already well-established that gene expression negatively correlates with methylation levels. Citing relevant literature to support this correlation would suffice.
5. The font size in Figure 4 is too small, making it difficult for readers to interpret the data. Increasing the font size would improve clarity.
Author Response
Response to reviewer 1 comments
By analyzing microarray-based transcriptomic and methylation data, the authors identified eighteen multi-evidence genes (MEGs), 12 of which are novel in RA research. Bioinformatics analysis revealed these genes are part of protein-protein interaction clusters linked to RA-related pathways and interact directly with RA-associated miRNAs, underscoring their potential role in disease progression. The manuscript contains significant design flaws. Below are my comments.
- The authors' use of "the highest number of common genes" as a criterion for data collection is not well-justified. The methylation dataset (GSE121192) used in the manuscript includes 46 samples across three different cell types, a detail the authors did not address. Given the small sample size (13 patients vs. 9 controls; 10 patients vs. 6 controls), the study's statistical power to identify meaningful genes is limited. Larger datasets, such as GSE15573 and GSE176168, could have been considered to enhance the robustness of the findings. Even though all the datasets involve small sample sizes, methods like meta-analysis could have been employed to improve detection power. Relying on one or two datasets is not an optimal strategy.
We thank the reviewer for this comment. We chose the two datasets based on specific selection criteria, which are now more clearly outlined in Section 4.1 of the revised manuscript in line 323-329. While we acknowledge the reviewer's suggestion to incorporate additional datasets, it is important to note that other datasets, such as GSE176168, did not meet our inclusion criteria. For instance, GSE176168 includes patients undergoing anti-TNF therapy, and we intentionally excluded studies involving any form of treatment to avoid confounding factors that could influence gene expression and methylation patterns.
Additionally, the combination of two omics modalities—methylation and gene expression—provides a more comprehensive understanding of RA pathogenesis, strengthening our findings. While larger datasets could enhance statistical power, we believe the data we used is sufficient to draw meaningful conclusions about key molecular markers in RA, since it is derived from (as rightly pointed out by the reviewer), 13 RA patients for one set and 10 RA patients for the other which we believe gives a fair idea of the key markers involved in RA pathologies.
- Have the authors reviewed the original papers that provided the publicly available datasets? The number of differentially expressed genes (DEGs) reported in this manuscript appears to differ significantly from those identified in the original studies. The authors should provide a clear explanation for these discrepancies.
We have indeed reviewed the original papers, however, a common practice for multiomics data integration is to normalize the data as per a certain adjusted p-value and log FC. To elaborate, the discrepancy is observed as a result of the data filters applied during data processing and analysis. In the original study for GSE121192, a p-value (0.0003<p<0.001) and log FC= 1.5 was applied whereas, for GSE56649, an adjusted p-value less than 0.05 and log FC 1.5 was applied. During data integration, we used an adjusted p-value ≤ 0.05 and log FC =1 for both the datasets to normalize the data for data integration.
- The manuscript lacks substantial discoveries or significant findings. Conducting several bioinformatics analyses alone is insufficient for a publication-level scientific paper. Experimental validation of the identified genes is needed to lend greater credibility to the results.
While wet lab validation provides crucial insights into real-life phenomena, computational tools are increasingly employed to streamline these experiments, especially in biomarker discovery. The aim of this study was to utilize computational methods to identify candidate markers across multiple omics levels for the treatment of rheumatoid arthritis (RA). Our findings provide substantial evidence of the involvement of the selected markers in RA. Additionally we have added a few sentences in the discussion section to indicate the next step would be to test the selected candidate markers in wet lab experiments.
- Figure 1 is essential, but it is already well-established that gene expression negatively correlates with methylation levels. Citing relevant literature to support this correlation would suffice.
Figure 1 refers to the eighteen genes identified in the study from the intersection of selected datasets. There have been many studies supporting that gene expression and methylation are negatively related but figure 1 is provided in context of dataset used in the study only. We also added a reference (line 129, reference 21) justifying the statement in the revised manuscript. In addition, we have added a new heatmap in the revised manuscript which is more informative and shows the expression levels of each gene indicated by a color key.
- The font size in Figure 4 is too small, making it difficult for readers to interpret the data. Increasing the font size would improve clarity.
Thank you for pointing out the issue with Figure 4's font size. We have increased the overall size and resolution of the figure to improve clarity. The adjustments made enhance the figure's readability significantly.

Reviewer 2 Report
Comments and Suggestions for Authors
The manuscript entitled ‘Integrated multi-omics approach for the identification of novel candidate therapeutic markers in Rheumatoid Arthritis’ written by Muhammad Hamza Tariq et al. presents interesting results regarding the identification of molecular biomarkers related to rheumatoid arthritis. The authors used two publicly available datasets to compare transcriptomic and epigenomic data between individuals with this disease and healthy controls. The authors identified twelve novel genes that have a biomarker character and potential to be used as therapeutic targets. Further bioinformatic analysis reveals associations with signaling pathways and biological functions. The miRNA-mRNA interactions were also explored and miRNAs regulating rheumatoid arthritis-related genes were shown.
Obtained results disclosed genes that altered regulation potentially contribute to the development of rheumatoid arthritis and could serve as a therapeutic markers. However, the performed study is based on the analysis of only publicly available datasets (the authors did not perform validation using biological material) with a low number of samples, which significantly reduces the importance of the presented work. Furthermore, the methodology and results were not adequately explained and detailed; therefore, some important improvements should be introduced to make the manuscript clear, scientifically sound, reproducible, and of greater importance to the scientific community. The English language is clear.
General concept comments
1. In the Introduction section, the mechanisms underlying rheumatoid arthritis, including signaling pathways, inflammatory mediators, or regulatory mechanisms (e.g. gene methylation, differentially expressed genes, miRNA regulation) should be described in more detail to help the readers familiarise themselves with processes involved in this disease, and to provide the complete background of this topic. Extended up-to-date overview can help to highlight gaps in the knowledge.
2. In the Abstract section, the authors stated that the cohorts were ‘matched’ (lines 23-24). However, in the rest of the text, there is no mention of matching cohorts. Please clarify how the compared groups were matched and what features (age, sex, or other?) were used.
3. The performed study involves statistical testing of a large number of genes. In such a scenario, a correction of p values for multiple testing is typically carried out using the Benjamini-Hochberg (or other) method. Please clarify whether such a correction was performed in the study or justify why such a procedure was not applied.
4. The presented work is grounded only on computational methods and the low number of samples in the analysed data; therefore, validation of the obtained results in biological samples is absolutely necessary to confirm the clinical importance of indicated markers; however, the presented results give a valuable preliminary view of the genetic markers of reumatoid arthritis.
5. The main limitations of the study should be mentioned in the Discussion section.
Specific comments
1. In paragraph 4.2, the authors stated ‘For data analysis, the GEO2R tool was utilised to create volcano plots’ (line 279), but the information about the tool was used for differential expression analysis and differential methylation analysis is missing. If it was also GEO2R, it should be specified. Furthermore, created volcano plots should be provided in the main text or in the Supplementary Material.
2. In lines 73-74, the authors stated ‘The comparative differential expression pattern of these eighteen common genes is presented in the form of a heatmap in Figure 1.’; however, there is no information before in the text on how these 18 genes were selected. Furthermore, the heatmap in Figure 1 is rather low informative. Instead of this figure, I suggest the authors to provide a table with genes symbols and names, p values, fold change values, and methylation status (hypo-, hypermethylation).
3. Figure 3 would be more readable if a legend for cluster node colours would be added to this figure, because the words ‘cluster’ in the graph are difficult to distinguish from gene symbols. Similarly, a similar legends for node colors and shapes can be added to Figures 5 and 6 to make the figures easier to understand and interpret by the readers.
4. A typo in line 303.
I believe that my suggestions will help the authors improve the quality of their manuscript.
Author Response
Response to reviewer 2 comments
The manuscript entitled ‘Integrated multi-omics approach for the identification of novel candidate therapeutic markers in Rheumatoid Arthritis’ written by Muhammad Hamza Tariq et al. presents interesting results regarding the identification of molecular biomarkers related to rheumatoid arthritis. The authors used two publicly available datasets to compare transcriptomic and epigenomic data between individuals with this disease and healthy controls. The authors identified twelve novel genes that have a biomarker character and potential to be used as therapeutic targets. Further bioinformatic analysis reveals associations with signaling pathways and biological functions. The miRNA-mRNA interactions were also explored and miRNAs regulating rheumatoid arthritis-related genes were shown.
Obtained results disclosed genes that altered regulation potentially contribute to the development of rheumatoid arthritis and could serve as a therapeutic markers. However, the performed study is based on the analysis of only publicly available datasets (the authors did not perform validation using biological material) with a low number of samples, which significantly reduces the importance of the presented work. Furthermore, the methodology and results were not adequately explained and detailed; therefore, some important improvements should be introduced to make the manuscript clear, scientifically sound, reproducible, and of greater importance to the scientific community. The English language is clear.
General concept comments
- In the Introduction section, the mechanisms underlying rheumatoid arthritis, including signaling pathways, inflammatory mediators, or regulatory mechanisms (e.g. gene methylation, differentially expressed genes, miRNA regulation) should be described in more detail to help the readers familiarize themselves with processes involved in this disease, and to provide the complete background of this topic. Extended up-to-date overview can help to highlight gaps in the knowledge.
We agree with the reviewer and have rewritten the introduction to incorporate the points raised by the reviewer. The changes can be seen from lines 38-96 and in particular the pathophysiology of RA is discussed from line 49-57.
- In the Abstract section, the authors stated that the cohorts were ‘matched’ (lines 23-24). However, in the rest of the text, there is no mention of matching cohorts. Please clarify how the compared groups were matched and what features (age, sex, or other?) were used.
We thank the reviewers for pointing out this error. The groups were matched based on the biological sample under investigation. Since RA samples can be derived from peripheral blood as well as synovial fluid, we matched patient cohorts to compare DEGs from peripheral blood only to avoid any bias resulting from confounding factors. Additionally, studies involving only drug naïve RA patients were selected. This is now stated more clearly in the revised manuscript under the methods section 4.1 as shown by yellow highlighted text in line 323-329.
- The performed study involves statistical testing of a large number of genes. In such a scenario, a correction of p values for multiple testing is typically carried out using the Benjamini-Hochberg (or other) method. Please clarify whether such a correction was performed in the study or justify why such a procedure was not applied.
We thank the reviewer for pointing this out. In fact, the analysis in our study was conducted using the GEO2R tool, which offers the option to apply the Benjamini & Hochberg method for adjusting p-values to control for false discovery rates. We selected this option during our analysis, and we have now emphasized this detail in the methods section of the revised manuscript for clarity from line 331-335.
- The presented work is grounded only on computational methods and the low number of samples in the analysed data; therefore, validation of the obtained results in biological samples is absolutely necessary to confirm the clinical importance of indicated markers; however, the presented results give a valuable preliminary view of the genetic markers of rheumatoid arthritis.
We thank the reviewer for their positive end note and would like to emphasize that while wet lab validation provides crucial insights into real-life phenomena, computational tools are increasingly employed to streamline these experiments, especially in biomarker discovery. The aim of this study was to utilize computational methods to identify candidate markers across multiple omics levels in RA. Our findings provide substantial evidence of the involvement of the selected markers in RA. Additionally we have added a few of sentences in the discussion section (line 314-318) to indicate the next step would be to test the selected candidate markers in wet lab experiment.
- The main limitations of the study should be mentioned in the Discussion section.
We have now added a new paragraph (highlighted in yellow) at the end of the discussion (from line 310-318) section to include limitations.
Specific comments
- In paragraph 4.2, the authors stated ‘For data analysis, the GEO2R tool was utilized to create volcano plots’ (line 279), but the information about the tool was used for differential expression analysis and differential methylation analysis is missing. If it was also GEO2R, it should be specified. Furthermore, created volcano plots should be provided in the main text or in the Supplementary Material.
We thank the reviewer for their suggestions and have now clarified the use of GEO2R tool in designing volcano plots as stated in line 331 and have now added the volcano plots as supplementary figure 1 in the revised manuscript.
- In lines 73-74, the authors stated ‘The comparative differential expression pattern of these eighteen common genes is presented in the form of a heatmap in Figure 1.’; however, there is no information before in the text on how these 18 genes were selected. Furthermore, the heatmap in Figure 1 is rather low informative. Instead of this figure, I suggest the authors to provide a table with genes symbols and names, p values, fold change values, and methylation status (hypo-, hypermethylation).
We thank the reviewer for their comment and suggestion. We agree and have made the following changes; 1. We have changed the order of the Figure 1 to highlight how the 18 genes were selected in the form of a venn diagram and then inserted the heat map,2) we have added a new heatmap that better displays the expression levels of the 18 genes along with a colour key and, 3) we have provided the table as per reviewers suggestion regarding the gene symbols, names and p-values for both transcriptomic and epigenomics data sets and this is indicated in the revised manuscript as supplementary table 1.
- Figure 3 would be more readable if a legend for cluster node colours would be added to this figure, because the words ‘cluster’ in the graph are difficult to distinguish from gene symbols. Similarly, a similar legends for node colors and shapes can be added to Figures 5 and 6 to make the figures easier to understand and interpret by the readers.
We agree and have updated the figure captions as suggested.
- A typo in line 303.
We thank the reviewer for pointing this out, the typo has been corrected.
I believe that my suggestions will help the authors improve the quality of their manuscript.
Thank you to the reviewer for the valuable suggestions, we have incorporated all the changes and we agree with the reviewer that the quality of the manuscript is now significantly improved.

Reviewer 3 Report
Comments and Suggestions for Authors
The authors must address several critical concerns.
1. A compelling, content-accurate title will entice researchers to delve deeper into the manuscript.
2. The abstract for this paper requires significant revision. It fails to clearly articulate the study's novelty or rationale. The objective section is overly wordy, lacking a concise summary of the overall justification. Similarly, the results section is incomplete, omitting crucial experimental data and findings necessary to support the conclusions.
3. The abstract conclusion should offer a concise, comprehensive summary of the study's key findings, moving beyond a mere restatement of the results. Rather than repeating sentences from the Results section, it should incorporate the authors' insights about the implications and broader significance of the research.
4. The current introduction provides only general, textbook-level information about rheumatoid arthritis (RA) biology that lacks relevance for this paper. Instead, the introduction should focus more narrowly on the specific bioinformatic methods and topics that will be explored throughout the paper in the context of RA treatment.
5. The concluding paragraph of the introduction should clearly communicate the paper's significance, its central objective, and pique the reader's interest in the study's methodology.
6. In the Results section, simply present the data objectively, without drawing any conclusions or making interpretations. The purpose of this section is to report the findings factually, not to analyze or infer meaning from them.
7. The Discussion section should offer a comprehensive interpretation of the findings, contextualizing their clinical relevance to metabolic syndrome and periodontitis. Additionally, it is important to acknowledge the limitations of the in vitro model and propose future in vivo validation studies.
8. While the paper lacks evidence of biological interactions among the candidate genes, targeted experiments could validate the bioinformatic findings. Such interventions, like blocking specific genes, microRNAs, or signaling pathways, would help confirm the causal role of identified mediators in the observed crosstalk.
9. This study lacks functional assays to evaluate how the cultured cells impact specific osteoblast or osteoclast functions. Incorporating such assays would offer a more comprehensive understanding of the underlying tissue dysfunction.
10. To evaluate the expression of candidate genes and miRNAs in patient samples, the authors should conduct quantitative RT-PCR (qRT-PCR) to analyze mRNA levels and Western blotting to assess protein levels.
11. The conclusion should offer a more comprehensive and impactful summary of the key findings, highlighting their broader implications and providing a stronger, more insightful closing to the paper. Similarly, the discussion section should delve deeper into analyzing the implications of the findings, considering the wider context and potential directions for future research.
Comments on the Quality of English LanguageThe manuscript requires substantial language and stylistic improvements to enhance clarity and coherence. Rephrasing key sentences would boost readability, so it is highly recommended to work with a native English editor or professional language services to ensure a polished, fluid narrative.
Author Response
Response to reviewer 3 comments
- A compelling, content-accurate title will entice researchers to delve deeper into the manuscript.
We agree with the reviewers comment regarding a more content accurate title and have changed the title to "Identification of Novel Therapeutic Biomarkers in Rheumatoid Arthritis: A Combined Bioinformatics and Integrated Multi-Omics Approach” in the revised manuscript
- The abstract for this paper requires significant revision. It fails to clearly articulate the study's novelty or rationale. The objective section is overly wordy, lacking a concise summary of the overall justification. Similarly, the results section is incomplete, omitting crucial experimental data and findings necessary to support the conclusions.
We have taken on board the reviewer’s suggestion and have re-written the abstract as indicated in lines 19-34.
- The abstract conclusion should offer a concise, comprehensive summary of the study's key findings, moving beyond a mere restatement of the results. Rather than repeating sentences from the Results section, it should incorporate the authors' insights about the implications and broader significance of the research.
We have taken on board the reviewer’s suggestion and have re-written the abstract.
- The current introduction provides only general, textbook-level information about rheumatoid arthritis (RA) biology that lacks relevance for this paper. Instead, the introduction should focus more narrowly on the specific bioinformatic methods and topics that will be explored throughout the paper in the context of RA treatment.
We agree and the introduction section has been updated as suggested.
- The concluding paragraph of the introduction should clearly communicate the paper's significance, its central objective, and pique the reader's interest in the study's methodology.
We agree and the introduction section has been updated as suggested
- In the Results section, simply present the data objectively, without drawing any conclusions or making interpretations. The purpose of this section is to report the findings factually, not to analyze or infer meaning from them.
We thank the reviewer for this feedback however, we feel that the way the results are currently written are concise and do not actually draw upon conclusions or infer any meaning from the results. The results are discussed appropriately in the discussion section.
- The Discussion section should offer a comprehensive interpretation of the findings, contextualizing their clinical relevance to metabolic syndrome and periodontitis. Additionally, it is important to acknowledge the limitations of the in vitro model and propose future in vivo validation studies.
We disagree with the reviewer’s suggestion and would like to highlight that there is no indication of metabolic syndrome or periodontitis throughout the whole manuscript. Including these is beyond the scope of this manuscript.
- While the paper lacks evidence of biological interactions among the candidate genes, targeted experiments could validate the bioinformatic findings. Such interventions, like blocking specific genes, microRNAs, or signaling pathways, would help confirm the causal role of identified mediators in the observed crosstalk.
This study is purely based on computational tools and takes advantage of the use of bioinformatics tools for biomarker discovery. While wet lab validation provides crucial insights into real-life phenomena, computational tools are increasingly employed to streamline these experiments, especially in biomarker discovery. The aim of this study was to utilize computational methods to identify candidate markers across multiple omics levels in RA. Our findings provide substantial evidence of the involvement of the selected markers in RA. Additionally we have added a few sentences in the discussion section (lines 311-318) to indicate the next step would be to test the selected candidate markers in wet lab experiments and changed the title of our paper to clearly indicate the biomarker discovery aspect of our work.
- This study lacks functional assays to evaluate how the cultured cells impact specific osteoblast or osteoclast functions. Incorporating such assays would offer a more comprehensive understanding of the underlying tissue dysfunction.
We disagree with the reviewer’s suggestion and such experiments utilizing osteoblasts and osteoclasts are not intended for testing the presence of therapeutic markers. Future work will include the use of synovial fibroblasts from RA patients to mimic the RA pathology at the cellular level and the presence and absence of the candidate markers. However, we have taken the reviewers comment on board and changed the title of the study to "Identification of Novel Therapeutic Biomarkers in Rheumatoid Arthritis: A Combined Bioinformatics and Integrated Multi-Omics Approach” in the revised manuscript, to clearly state that this study utilized computational tools and also added a new paragraph (highlighted in yellow) at the end of the discussion (from line 311-318) section to include limitations of our work.
- To evaluate the expression of candidate genes and miRNAs in patient samples, the authors should conduct quantitative RT-PCR (qRT-PCR) to analyze mRNA levels and Western blotting to assess protein levels.
Conducting clinical sample analysis is definitely planned for the future but the purpose of this study was to emphasize the importance of bioinformatics and multiomics in therapeutic markers discovery. As such no wet lab validation or clinical sample analysis was conducted for the current study. However, we addressed the reviewers comment by adding a few sentences in the discussion section (lines 311-318) to indicate the next step would be to test the selected candidate markers in wet lab experiments.
- The conclusion should offer a more comprehensive and impactful summary of the key findings, highlighting their broader implications and providing a stronger, more insightful closing to the paper. Similarly, the discussion section should delve deeper into analyzing the implications of the findings, considering the wider context and potential directions for future research.
We believe we already have provided a through discussion of our results; however, we have added a paragraph in the discussion section to further summarise our findings, added limitatons (lines 303-318) and have now revised the conclusion section to highlight the implications of our findings (lines 394-401).
Comments on the Quality of English Language
The manuscript requires substantial language and stylistic improvements to enhance clarity and coherence. Rephrasing key sentences would boost readability, so it is highly recommended to work with a native English editor or professional language services to ensure a polished, fluid narrative.
We thank the reviewer for this feedback and have rephrased several sentences in the revised manuscript to increase clarity and also proof read and language edited the whole manuscript.

Reviewer 4 Report
Comments and Suggestions for Authors
The manuscript investigates the integration of transcriptomics and epigenomics to identify MEGs (methylation-regulated genes) associated with rheumatoid arthritis (RA). The authors present novel findings regarding MEGs and their potential roles in RA pathogenesis.
Major drawbacks:
1. The results could be presented more effectively with the inclusion of figures or tables that summarize the findings. For example, a heatmap should ideally include a detailed color scale indicating the expression level of each gene in every sample, which would provide more precise information rather than just indicating higher or lower expression. A table that shows the characteristics of each dataset. A comprehensive figure shows the computational approach used in this study. Figures show the prediction potential of 18 intersecting genes between samples.
2. While the statistical methods are mentioned, a more detailed explanation of how significance was assessed would strengthen the analysis.
3. While the study identifies gene interactions and pathways, it doesn’t make a strong connection to how these findings could improve RA treatment or diagnosis.
4. The study only analyzes peripheral blood samples, which may not capture all of the significant changes that occur in other tissues directly impacted by rheumatoid arthritis (RA), such as synovial tissues.
5. The explanation of how the various data sets (transcriptomics and epigenomics) were combined and evaluated might be improved. It’s not clear how these analyses were combined or what specific criteria were used to identify the 18 common genes.
6. The abstract mentions a "multi-omics approach" but doesn't explain exactly how it was used.
7. There are various abbreviations (e.g., DMARDs, GWAS) that may be confusing to the reader. Either reduce the number of abbreviations or ensure they are necessary and properly explained.
8. The introduction mentions merging transcriptomic and epigenomic data, however, it does not explain how these data types can lead to the identification of new treatments or diagnostics.
9. The introduction establishes the context for adopting multi-omics, but it does not properly clarify the hypothesis or the researchers' goals with this technique. A sentence about the research goals and how they will address the knowledge gap would make the introduction stronger.
10. The introduction repeatedly mentions that the exact cause of RA is unclear. This point could be stated once, and the rest of the paragraph could focus more on the novel approach used in the current study.
11. The introduction mentions that remission remains challenging and that RA is not fully understood, but it doesn’t clearly state why existing research is lacking or what gaps specifically need to be filled. Highlighting the specific limitations of current treatments would provide a stronger rationale for the study.
12. Terms like "multi-omics" and "holistic approach" are used but not explained in enough detail for non-experts. More clarification on what "multi-omics" means in this context would make the text clearer.
13. The introduction jumps between different topics (prevalence, symptoms, genetics, treatments) without smoothly connecting them. A more focused flow, starting with prevalence, then symptoms, and moving towards the need for multi-omics, would help the reader follow the argument better.
Comments on the Quality of English LanguageThe manuscript contains several grammatical errors and awkward phrasing that hinder readability. A thorough proofreading is necessary to improve clarity.
Author Response
Response to reviewer 4 comments
The manuscript investigates the integration of transcriptomics and epigenomics to identify MEGs (methylation-regulated genes) associated with rheumatoid arthritis (RA). The authors present novel findings regarding MEGs and their potential roles in RA pathogenesis.
Major drawbacks:
- The results could be presented more effectively with the inclusion of figures or tables that summarize the findings. For example, a heatmap should ideally include a detailed color scale indicating the expression level of each gene in every sample, which would provide more precise information rather than just indicating higher or lower expression. A table that shows the characteristics of each dataset. A comprehensive figure shows the computational approach used in this study. Figures show the prediction potential of 18 intersecting genes between samples.
We agree with the reviewer’s suggestions and have now updated the heatmap to include all the necessary information and provided a supplementary table (supplementary table 1) indicating the gene symbols, names and p-values for both transcriptomic and epigenomics genes as well as include a schematic to show the methodology used in the study. This is presented as Figure 7 in the revised manuscript.
- While the statistical methods are mentioned, a more detailed explanation of how significance was assessed would strengthen the analysis.
We agree with the reviewer and have now updated the method section to include the detailed method of analyzing and processing the data with additional information stating Benjamini & Hochberg (False discovery rate) method was used to apply adjustments. We used adjusted p-values and log transformation method to filter out the genes. This is now highlighted in lines 331-336 of the revised manuscript.
- While the study identifies gene interactions and pathways, it doesn’t make a strong connection to how these findings could improve RA treatment or diagnosis.
Genes identified to be involved in RA related gene ontologies and KEGG pathways can be considered as potential biomarkers for early diagnosis and observing disease development. Similarly, as expression of these identified genes are altered in RA, hence, they could be considered as potential targets while designing treatment options for RA. Many previous studies have been conducted to identify potential biomarkers for RA using same bioinformatics-based strategies, for example CCL5 was identified as a potential biomarker for RA diagnosis using bioinformatics based strategies and was subsequently shown to be differentially expressed in RA patients[i], validating the precision of opted methodology in our study. We have now added this information in the discussion section in lines 303-310.
- The study only analyzes peripheral blood samples, which may not capture all of the significant changes that occur in other tissues directly impacted by rheumatoid arthritis (RA), such as synovial tissues.
We agree with the reviewer that considering other biological samples such as synovial fluid or tissue would shed more light on molecular aspect of the disease however, for the purposes of avoiding any tissue/sample related bias, we only included studies using peripheral blood as it is a well-established and validated source for identifying biomarkers and molecular changes associated with disease.
Moreover, blood samples provide critical insights into systemic inflammation and immune dysregulation, which are key features of RA. Future studies could certainly explore the relationship between peripheral blood changes and those observed in synovial tissues, but our current study’s design was intentionally structured to provide robust, unbiased data from a practical and widely applicable sample type. We have added this information in lines 311-318 of the revised manuscript.
- The explanation of how the various data sets (transcriptomics and epigenomics) were combined and evaluated might be improved. It’s not clear how these analyses were combined or what specific criteria were used to identify the 18 common genes.
The presentation of the results, particularly regarding data comparison and heat map generation, has been updated in the revised manuscript. We have rearranged Figures 1 and 2 to enhance clarity. This new sequence clearly demonstrates that the Venn analysis was conducted first, followed by the heat map analysis of the 18 common genes identified.
- The abstract mentions a "multi-omics approach" but doesn't explain exactly how it was used.
We agree and have updated this in the introduction, methods and results section of the revised manuscript to state that the study used two omics modalities and integrated data from transcriptomics and epigenomics data sets.
- There are various abbreviations (e.g., DMARDs, GWAS) that may be confusing to the reader. Either reduce the number of abbreviations or ensure they are necessary and properly explained.
The abbreviations used in this paper were previously defined in their full forms, although they were not repeatedly used thereafter. However, we chose to include them because these abbreviations (e.g., DMARs and GWAS) are commonly recognized within the field, which facilitates readability for experts familiar with these terms.
- The introduction mentions merging transcriptomic and epigenomic data, however, it does not explain how these data types can lead to the identification of new treatments or diagnostics.
We agree and the introduction section has been updated, as suggested and now clearly states how data integration from multiomics can lead to biomarker discovery. This information is mentioned in lines 78-87 of the revised manuscript.
- The introduction establishes the context for adopting multi-omics, but it does not properly clarify the hypothesis or the researchers' goals with this technique. A sentence about the research goals and how they will address the knowledge gap would make the introduction stronger.
We agree and the introduction section has been updated, as suggested. We believe the revised introduction now makes our goals and study objectives stand out.
- The introduction repeatedly mentions that the exact cause of RA is unclear. This point could be stated once, and the rest of the paragraph could focus more on the novel approach used in the current study.
We agree and the introduction section has been updated, as suggested.
- The introduction mentions that remission remains challenging and that RA is not fully understood, but it doesn’t clearly state why existing research is lacking or what gaps specifically need to be filled. Highlighting the specific limitations of current treatments would provide a stronger rationale for the study.
We agree and the introduction section has been updated, as suggested.
- Terms like "multi-omics" and "holistic approach" are used but not explained in enough detail for non-experts. More clarification on what "multi-omics" means in this context would make the text clearer.
We agree and the introduction section has been updated, as suggested. This information can be seen in lines 80-87 of the revised manuscript.
- The introduction jumps between different topics (prevalence, symptoms, genetics, treatments) without smoothly connecting them. A more focused flow, starting with prevalence, then symptoms, and moving towards the need for multi-omics, would help the reader follow the argument better.
We agree and the introduction section has been updated, as suggested.
Comments on the Quality of English Language
The manuscript contains several grammatical errors and awkward phrasing that hinder readability. A thorough proofreading is necessary to improve clarity.
We agree and we have now carefully proofread the content to ensure it meets high standards of language and readability.

Round 2
Reviewer 1 Report
Comments and Suggestions for Authors
The authors have made their best efforts to address my concerns from the previous round of review. I have no more comments.
Author Response
Reviewers comment: The authors have made their best efforts to address my concerns from the previous round of review. I have no more comments.
Reply: we thank the reviewers for their valuable time in reviewing our manuscript. Their comments definitely improved the overall quality of the manuscript.
Reviewer 2 Report
Comments and Suggestions for Authors
I appreciate the efforts of the authors to respond to my comments. Some of the figures go beyond the document area, and their size must be reduced to be fully visible. Although this work is not impressive, the presented results provide a valuable preliminary view of the promising circulatory biomarkers of rheumatoid arthritis. I have no further comments.
Author Response
Comment: I appreciate the efforts of the authors to respond to my comments. Some figures go beyond the document area, and their size must be reduced to be fully visible. Although this work is not impressive, the presented results provide a valuable preliminary view of the promising circulatory biomarkers of rheumatoid arthritis. I have no further comments.
Reply: We thank the reviewer for their valuable time reviewing our manuscript. Their comments improved the overall quality of our manuscript. Regarding the reviewer's concern about the figures' size, we would draw their attention to the zip file containing all figures in the desired quality and size. The size of the figures appears oversized in the embedded template for some unknown reason.
Reviewer 3 Report
Comments and Suggestions for Authors
The research motive is unclear, and the hypothesis lacks academic validity. Additionally, the data does not adequately address the research question. The conclusions drawn are inaccurate, as they are based on unsupported assumptions.
Author Response
Comment: The research motive is unclear, and the hypothesis lacks academic validity. Additionally, the data does not adequately address the research question. The conclusions drawn are inaccurate, as they are based on unsupported assumptions.
Reply: We respectfully disagree with your assessment that the research motive is unclear and the hypothesis lacks academic validity. In the first round of revisions, we addressed these points thoroughly by refining the research objectives and changing the title based on your earlier suggestions. We believe the current version presents a clear and academically robust rationale for the study.
Similarly, regarding your concern about the data and conclusions, we carefully incorporated the feedback from the initial round to ensure the data aligns with the research question and supports our findings. We are confident that the revised manuscript reflects these improvements.
We also noticed that some concerns raised in this round were not highlighted during the initial review. While we are happy to address any remaining issues, we kindly request clarification to understand better how we can meet your expectations and improve the manuscript further.
Reviewer 4 Report
Comments and Suggestions for Authors
The authors have addressed some of the previous comments; however, several major issues remain unresolved, particularly concerning the functional enrichment analysis and the methodological rigor in data interpretation
1. The authors did not account for the False Discovery Rate (FDR) in their enrichment analysis using DAVID. Without FDR correction, the significance of the enriched biological functions may be unreliable, potentially leading to misleading conclusions. I strongly recommend that the authors apply FDR correction to ensure the validity and robustness of the findings .
2. It appears the authors may not be fully familiar with the correct methodology for constructing a heatmap. The heatmap should display all samples, appropriately grouped into RA patients and healthy controls. This would result in a total of 64 samples for the 18 genes of interest. I recommend revising the heatmap to properly represent the samples and their corresponding groupings.
3. In my previous comments, I requested the inclusion of ROC curves to assess the potential of the 18 genes as predictive biomarkers. This analysis is crucial for evaluating the predictive power of the identified genes. I urge the authors to include ROC curves for all 18 genes to strengthen the conclusions regarding their predictive potential.
4. I would like to see a more detailed presentation of the biological function and KEGG pathway enrichment results for the 18 genes and their associated clusters, as extracted from DAVID. This information should be provided in an Excel file for greater clarity and transparency. Additionally, the details of the DEGs (differentially expressed genes) should be presented in a separate Excel file. These files should include the full set of genes identified in the analysis, along with associated statistical measures (e.g., p-values, fold changes), and any relevant annotations.
5. In Section 2.2, the authors state that an absolute log fold change (log FC) of 1 was used for both datasets, but the volcano plot does not appear to support this claim. Specifically, the Y-axis does not correspond to the expected threshold of 0.05, raising concerns about the validity of the results. Furthermore, the authors have used p-values as the criterion for identifying differentially expressed genes (DEGs), rather than the more robust FDR, which is commonly preferred for controlling false positives in differential expression analyses. This raises doubts about the reliability of the results. I strongly recommend that the authors clarify why p-values were used instead of FDR. Additionally, the volcano plot should be included in the manuscript, clearly showing the absolute log FC and p-values, and highlighting the common genes identified across datasets, as well as some of the top DEGs. This would greatly enhance the readability and interpretation of the results.
6. The KEGG pathways shown in Figure 5 appear to be selected based on p-values. If this is indeed the case, I suggest the authors clarify this point. Furthermore, I recommend that only pathways with an FDR value <0.05 be considered to ensure the selection of truly significant pathways.
Comments on the Quality of English LanguageSome areas of the manuscript still exhibit unclear phrasing and grammatical errors that need to be addressed for clarity and readability.
Author Response
- The authors did not account for the False Discovery Rate (FDR) in their enrichment analysis using DAVID. Without FDR correction, the significance of the enriched biological functions may be unreliable, potentially leading to misleading conclusions. I strongly recommend that the authors apply FDR correction to ensure the validity and robustness of the findings.
We thank the reviewer for this important suggestion. We have applied the FDR correction in the revised version which highlights bubble plots with FDR corrected values in the modified Figure 4.
- It appears the authors may not be fully familiar with the correct methodology for constructing a heatmap. The heatmap should display all samples, appropriately grouped into RA patients and healthy controls. This would result in a total of 64 samples for the 18 genes of interest. I recommend revising the heatmap to properly represent the samples and their corresponding groupings.
We thank the reviewer for pointing this out and upon reflection, we recognized that the original presentation of the heatmap did not fully capture our findings. Following the reviewer’s suggestion, we have restructured the heatmaps to display all samples from each study, along with the relative expression levels of the key genes. These changes are highlighted in Figure 2 of the revised manuscript.
- In my previous comments, I requested the inclusion of ROC curves to assess the potential of the 18 genes as predictive biomarkers. This analysis is crucial for evaluating the predictive power of the identified genes. I urge the authors to include ROC curves for all 18 genes to strengthen the conclusions regarding their predictive potential.
We appreciate the reviewer’s concern regarding the predictive power of the identified genes. However, we believe that the inclusion of ROC curves, while valuable, would be more appropriate as part of future work. The primary aim of the current study was to identify common gene candidates across the two omics levels, which we have successfully done, identifying 18 genes as common across the datasets. We have conducted a comprehensive analysis of these genes to demonstrate their potential relevance in RA, and we believe the current findings are sufficient to support our conclusions. Although ROC curve analysis would undoubtedly strengthen the study, it is beyond the scope of this particular work.
4.I would like to see a more detailed presentation of the biological function and KEGG pathway enrichment results for the 18 genes and their associated clusters, as extracted from DAVID. This information should be provided in an Excel file for greater clarity and transparency. Additionally, the details of the DEGs (differentially expressed genes) should be presented in a separate Excel file. These files should include the full set of genes identified in the analysis, along with associated statistical measures (e.g., p-values, fold changes), and any relevant annotations.
In the revised manuscript, we have included a supplementary file that provides detailed information on the DEGs, including p-values and logFC values (Supplementary File 1 and Supplementary File 2). Additionally, the revised version now includes Supplementary File 3, which contains the gene ontology details for all three clusters, as well as the DAVID-retrieved KEGG pathways in which the 18 genes are implicated.
- In Section 2.2, the authors state that an absolute log fold change (log FC) of 1 was used for both datasets, but the volcano plot does not appear to support this claim. Specifically, the Y-axis does not correspond to the expected threshold of 0.05, raising concerns about the validity of the results. Furthermore, the authors have used p-values as the criterion for identifying differentially expressed genes (DEGs), rather than the more robust FDR, which is commonly preferred for controlling false positives in differential expression analyses. This raises doubts about the reliability of the results. I strongly recommend that the authors clarify why p-values were used instead of FDR. Additionally, the volcano plot should be included in the manuscript, clearly showing the absolute log FC and p-values, and highlighting the common genes identified across datasets, as well as some of the top DEGs. This would greatly enhance the readability and interpretation of the results.
We thank the reviewer for pointing out this error to us. We have now reanalyzed our data and replotted the volcano plots as suggested by the reviewer. These changes are highlighted in Figure 1 (b) of the revised manuscript. With regards to the p-value, in our approach, we chose to use p-values as the initial criterion for DEG identification, as it provided a straightforward and interpretable measure of statistical significance for this exploratory analysis. Furthermore, we have already conducted robust analyses to assess the biological relevance of the common genes using FDR correction. Therefore, the presented results are highly reliable. We believe with the inclusion of new volcano plots and heatmaps, as suggested by the reviewer, the clarity of our data has been further enhanced and appears to be more streamlined.
6.The KEGG pathways shown in Figure 5 appear to be selected based on p-values. If this is indeed the case, I suggest the authors clarify this point. Furthermore, I recommend that only pathways with an FDR value <0.05 be considered to ensure the selection of truly significant pathways.
We would like to clarify that the representation in Figure 5 is not based on pathway enrichment analysis. Instead, it focuses on mapping the identified genes to KEGG pathways related to RA pathogenesis, with the aim of suggesting their potential role as biomarkers for RA. Collectively the 18 candidate genes did not show enrichment for any RA-related KEGG pathway, we chose to emphasize the connection between individual genes and KEGG pathways rather than presenting enrichment data. We hope this clarification adequately addresses your concern and strengthens the interpretation of our findings.

Round 3
Reviewer 4 Report
Comments and Suggestions for Authors
The revisions have significantly improved the manuscript, and I have no further concerns.